# Does highlighting COVID-19 disparities reduce or increase vaccine intentions? evidence from a survey experiment in a diverse sample in New York State prior to vaccine roll-out

Ashley Fox[1]*, Yongjin Choi[1,2,3], Heather Lanthorn[4], Kevin Croke[5]

1 Department of Public Administration and Policy, Rockefeller College of Public Affairs and Policy, University at Albany, State University of New York, Albany, NY, United States of America, 2 Vaccine Confidence Project, London School of Hygiene and Tropical Medicine, London, United Kingdom, 3 Laboratory of Data Discovery for Health, The University of Hong Kong, Hong Kong, China, 4 IDinsight, Asheville, North Carolina, United States of America, 5 Department of Global Health and Population, Harvard T.H. Chan School of Public Health, Boston, MA, United States of America

* afox3@albany.edu

**Data Availability Statement:** We will deposit the data in Harvard Dataverse upon publication of the manuscript. In addition, all replication materials will

## Abstract

Racial identity and political partisanship have emerged as two important social correlates of hesitancy towards COVID-19 vaccines in the United States. To examine the relationship of these factors with respondents' intention to vaccinate before the vaccine was available (November/December, 2020), we employed a multi-method approach: a survey experiment that randomized a vaccine-promotion message focused on racial equity in vaccine targeting, stepwise regression to identify predictors of hesitancy, and qualitative analysis of open-ended survey questions that capture how respondents reason about vaccination intentions. Experimental manipulation of a racial equity vaccine promotion message via an online survey experiment had no effect on intention-to-vaccinate in the full sample or in racial, ethnic and partisan subsamples. Descriptively, we find heightened hesitancy among non-Hispanic Black respondents (OR = 1.82, p<0.01), Hispanics (OR = 1.37, p<0.05), Trump voters (OR = 1.74, p<0.01) and non-Voters/vote Other (OR = 1.50, p<0.01) compared with non-Hispanic White respondents and Biden voters. Lower trust in institutions, individualism and alternative media use accounted for heightened hesitancy in Trump voters, but not non-Hispanic Blacks and Hispanics. Older age and female gender identity also persistently predicted lower vaccine intentions. Qualitatively, we find that most hesitant responders wanted to 'wait-and-see,' driven by generalized concerns about the speed of vaccine development, and potential vaccine side-effects, but little mention of conspiracy theories. Identity appears to be an important driver of vaccinate hesitancy that is not fully explained by underlying socioeconomic or attitudinal factors; furthermore, hesitancy was not significantly affected by racial equity messages in this setting.

be made fully available on a GitHub page at https://github.com/TheYongjinChoi/NYS-COVID19-Disparities-Survey.

**Funding:** AMF received funding for the data collection of this project from the Community Engaged COVID-19 Health Disparities Researcher group at the University at Albany. This was an internal university grant made possible through funding from the State of New York. The funders had no role in study design, data collection and analysis, decision to publish, or preparation of the manuscript.

**Competing interests:** The authors have declared that no competing interests exist.

## Introduction

As vaccinations rates for COVID-19 remain below the targets set by public health authorities in the U.S., it is important to understand both population-level willingness to vaccinate against COVID-19, as well as the reasons behind stated attitudes of vaccine acceptance or hesitancy in order to tailor interventions to increase uptake. Vaccine hesitancy, which has been defined as either a delay in acceptance *or* refusal of vaccination despite accessibility and affordability of vaccination services [1], remains a challenge in the United States: A recent review of the 106 articles from nationally representative samples found an overall COVID-19 vaccine acceptance rate ranging from 53.6% to 84.4% [2], and the population vaccination rate as of April 2022 was 66% [3]. As many COVID-19 related restrictions began to be lifted in early 2022, including vaccine mandates, many public health experts have warned about the further dampening effects on vaccine uptake, and the concurrent need for more informational campaigns to continue to encourage people to get vaccinated and boosted [4,5].

Even before COVID-19 vaccines became widely available to the general public, academic researchers and media reporters sought to identify subpopulations that expressed higher levels of vaccine hesitancy and lower intention to vaccinate against COVID-19. A review of 13 published studies with national samples in early 2021 found that African-Americans had significantly heightened hesitancy compared with other groups [6,7]. Further reinforcing these findings, in their review of 106 studies, Wang and Lie found that compared to non-Hispanic whites, identifying as non-Hispanic Black or Hispanic was associated with 2 to 6-fold higher chance of being COVID-19 vaccine hesitant, respectively [2]. However, as the vaccine roll-out began in early 2021, emerging polls and media content began to show that vaccine hesitancy has was declining in non-Hispanic Blacks, but remaining elevated in White Republicans. By March 2021polls showed that nearly 60% of white Republicans reported they would either not take the vaccine or were unsure if they would [8]. Research also increasingly shows supporters of former President Donald Trump to be particularly reluctant to vaccinate, despite the fact that Mr. Trump himself received the vaccine, and that his administration pushed for the rapid development of a COVID-19 vaccine [9,10]. This finding of elevated vaccine hesitancy in Republicans is consistent with broader polarization of views about COVID-19 along partisan lines, which emerged as early as the first weeks of lockdown in the US in March 2020 [11]. In a systematic review, Wang and Lie found at least 10 studies in which authors reported positive associations between identifying as a Republican or a conservative and intending to refuse vaccination [2].

To better understand these patterns, researchers have examined the sources of distrust in different groups, in order to unpack the mechanisms or attitudes which underly this distrust. A large literature examines broader vaccine hesitancy among African-Americans, rooting this mistrust of the medical establishment both in historical and contemporary practices and experiences [12–15]. While historic events have led to skepticism among African-Americans toward medical institutions in general and vaccines in particular, the roots of mistrust are not strictly historical. Overt racism and explicitly unethical practice has given way to less overt, implicit forms of racial bias and disregard in the health system [16,17]. Historic and on-going lived experience manifest as lower trust in medical research, scientists and doctors among non-Hispanic Blacks relative to other race-ethnic groups [16,18]. Less is known about vaccine hesitancy among Hispanic populations and its intersections with political partisanship, as party identification among Hispanics tends to vary by national origin [19]. However, a recent review of 13 studies found heightening COVID-19 vaccine hesitancy among Hispanics compared with non-Hispanics Whites, though to a lesser extent than non-Hispanic Blacks [6].

While medical mistrust has been identified as a source of hesitancy among non-Hispanic Blacks, less is know about how mistrust might affect intention to vaccinate among other

groups. Mistrust of government has been growing generally in American society for some time [20]. Growing mistrust in mainstream institutions has also been identified as an important predictor of support for former President Trump in the U.S. [21–23], although it also long predated the Trump administration and has fueled populist movements globally [24]. Furthermore, there is evidence that far-right extremist groups that spread false information about the 2020 US presidential election have continued similar activities regarding vaccines [25,26]. Studies find that certain sub-groups of political conservatives are more likely to be vaccine-hesitant and vulnerable to misinformation than others [27]. Political populism has also been tied to both ethno-nationalism (nativist and nationalistic tendencies) and a preference for "common sense" knowledge over elite knowledge, which contributes to skepticism in medical and scientific experts [28]. This suggests that low levels of trust in institutions–whether driven ascriptive characteristics or political partisanship–may be an important underlying driver of vaccine hesitancy.

The evolution of these partisan dynamics presents a dilemma for message framing aimed at increasing vaccine uptake for COVID-19 and beyond. On the one hand, to increase vaccine confidence among non-Hispanic Blacks, health communication scholars have recommended counteracting mistrust by acknowledging that medical mistrust is justifiable based on past and present harms [29]. This may also involve "role-modeling" from public officials, health authorities and influencers to help in building public trust as well as race concordance with the messenger [30].

On the other hand, it is also possible that framing COVID-19 as a disease that primarily affects minorities and/or prioritizing minority communities for vaccination may stimulate racial animus in non-Hispanic Whites, and thereby reduce their own intentions to vaccinate, especially among Whites with higher levels of racial bias. Thus, framing vaccine messages around race potentially could be a double-edged sword, as shown by several recent studies. Harell and Lieberman demonstrate that informing White Americans about racial disparities in COVID-19 mortality rates decreased support for public health measures among those with racially biased views, while increasing support among those with less bias [31]. Similarly, Skinner-Dorkenoo et al. found that manipulating exposure to information about COVID-19 racial disparities in infections and deaths reduced fear of COVID-19, empathy for those vulnerable to COVID-19, and support for safety precautions among those randomized to read about persistent inequalities [32]. Acknowledging racism and suggesting racial prioritization could build vaccine confidence in racial and ethnic minorities, but it could also raise resentment and reduce confidence among Whites.

In this context, we examine racial, ethnic and partisan differences in vaccine hesitancy, particularly the contribution of mistrust in institutions on intention to vaccinate, through a multi-method approach. Given variation in trust in the medical establishment and the novel context of a pandemic and new vaccine, we explore three key questions. First, does experimental exposure to a news report favoring minority prioritization for the vaccine, due to the disproportionate burden of COVID-19 on these groups, affect intent-to-vaccinate? If so, does the impact of reading this news report vary along race-ethnic and partisan lines? Second, how do respondents explain their intent-to-vaccinate (or not) in their own words? Do these explanations vary across race-ethnic and partisan lines? Third and finally, to what extent are race-ethnicity and party identification associated with intent-to-vaccinate for COVID-19 and what factors, including mistrust, are correlated with elevated hesitancy in these groups?

We studied these questions with data gathered from an online survey and experiment with a diverse sample of 1,353 respondents in New York State. New York State is an important context in which to study race-ethnic disparities in vaccine hesitancy as the New York City metro area was the epicenter of the first US wave of COVID-19, and non-Hispanic Black and

Hispanic populations, which are concentrated in New York City, were disproportionately affected by COVID-19 morbidity and mortality. However, NYS is also demographically and politically diverse; much of upstate is more rural, white and conservative than highly diverse communities in New York City and other locations downstate. We utilized linked closed-ended and open-ended questions to generate deeper insights on respondents' intentions to get a COVID-19 vaccine before the vaccine was available.

## Methods

### Sampling and administration

An online, self-administered survey was fielded between November 23-December 8, 2020—shortly before the first COVID-19 vaccine was approved in the US. This survey, implemented by Qualtrics, was only open to residents of New York State (NYS). Non-Hispanic Blacks and Hispanics were intentionally over-sampled, which allowed us to test for heterogenous effects of the survey experiment by race and ethnicity.

This study was approved by the Institutional Review Board of the University at Albany (IRB study number, 20X258). Respondents were informed that a university was fielding the survey. All survey participants were shown a study information form about their rights as research participants at the time they accessed the study and were required to affirmatively indicate consent in order to proceed. Upon completion they were shown a debriefing message that explained that there was deception involved (see Appendices for consent and debrief messages).

The median respondent took 23.2 minutes to complete the survey. Respondents who did not meet quality standards, *i.e.*, by completing the survey in an implausibly short time, were excluded from our analytic sample. The completion rate of the survey was 59% (59 completes for every 100 entrants).

The final sample of 1,353 respondents was intentionally stratified on race-ethnicity, with 429 non-Hispanic Whites, 443 non-Hispanic Blacks, and 481 Hispanics. Respondents were drawn from both Downstate (43%) from Upstate (57%). As a non-probability sample the results cannot be interpreted as representative of the population of the state of New York. Of the 1,353 respondents in our quantitative sample, 1,103 respondents (81.5%) provided at least one comprehensible answer to open-ended, free text questions meant to elicit qualitative data. These 1,103 respondents form our qualitative sample.

### Measurement

The survey included questions about the respondent's intention to receive the COVID-19 vaccination, trust in institutions, and a variety of demographic variables, including race and ethnicity (see Supplementary Materials 5 for the full questionnaire).

**Intent-to-vaccinate.** The primary dependent variable comes from the four-point survey item: "If a vaccine to prevent COVID-19 was approved by the FDA through normal procedures and available today for free to the public, would you. . . definitely get, probably get, probably not get, definitely not get. . . the vaccine as soon as possible?" While there are other ways to measure vaccine hesitancy that get more at underlying attitudes towards vaccines [33,34], our primary interest in this stidy was in understanding respondents stated intent to vaccinate. We analyzed intent to vaccinate in two ways. First, we used an ordered logit approach including all response categories, ordered from least to most hesitant. Second, we dichotomized the response with respondents who stated that they would definitely not get the vaccine as soon as possible set equal to 1 and all other respondents equal to 0. We report the results from the

ordered logistic regression analysis as main results and the results from the logistic regression in Supplementary Materials 3.

**Qualitative open-ended questions.**    Immediately following the close-ended question about vaccination intention an open-ended question was presented in which the respondent was either asked "briefly describe why you would get the vaccine as soon as possible" or briefly describe why you would not get the vaccine as soon as possible," based on whether they stated they likely would or would not vaccinate. The respondent could then record a free text answer explaining why they selected their chosen scale value.

**Race-ethnicity.**    To classify respondents' race-ethnicity, we use the approach taken by the US census, asking respondents about their Hispanic ethnicity first, followed by their selected racial classification. We then categorized respondents into mutually exclusive categories of non-Hispanic White, non-Hispanic Black, and people of Hispanic ethnicity. A small number of respondents categorized themselves as "Other" (n = 31). For the purpose of this analysis, this small group was pooled (with non-Hispanic Whites) to form the reference category in all regression analyses. We also ran the models with the 31 "Other" respondents excluded and the results were the same.

**Vote choice in 2020.**    Respondents were asked both about their party identification (Republican, Democrat, Independent, Other) and who they voted for in the 2020 Presidential election (President Biden, former President Trump, someone else, or did not vote). We created variables indicating whether the respondent reported that they voted for Trump (14%) in 2020, did not vote/voted for someone else (24%), or voted for Biden (60%). Previous surveys have found higher hesitancy among White Republicans [8], whereas other surveys have found hesitancy to be higher among Trump supporters specifically [35]. As self-reported presidential vote choice in 2020 and party identification are highly correlated, we present the results from the models with 2020 vote choice as the primary marker of political views as our main results. We include the models with party-identification as an independent variable in Supplementary Materials 3. We did not interact race and ethnicity with vote choice because the cell size becomes too small, particularly for Non-Hispanic Black Trump voters. However, the two are not perfectly co-linear- see Supplementary Materials 2 (S2 Table in S2 File).

## Covariates/Confounders

**Demographics.**    We adjusted for basic demographic characteristics including sex, age, education and household income. Additional information on question wording is available in Supplementary Materials 5.

**Personal experience with COVID-19.**    We control for a dichotomous measure equal to 1 where a respondent reported having a family member/relative who had COVID-19 or died from COVID-19 and 0 otherwise.

**Co-morbidities.**    We control for a composite measure of self-reported health conditions and risk factors including diabetes, heart disease, cancer, asthma, obesity and smoking. These were the best understood risk factors for severe COVID-19 and death at the time.

**Religiosity.**    We controlled for a dichotomous measure of religiosity using an indicator variable for respondents who indicated that religion is "very important" in their life.

## Mediators of vaccine hesitancy/confidence

**Mis/trust of medico-pharmaceutical institutions.**    We asked respondents "how much confidence do you have that that the following different political institutions would act in the best interest of the public when it comes to researching, developing, and distributing a COVID-19 vaccine?": federal government, federal health agencies (*e.g.*, the FDA and CDC),

local health departments, medical scientists and researchers, US pharmaceutical companies, and physicians. Trust was measured on a 5-point Likert scale where a higher number represents greater trust. We averaged across the responses to create a composite score of distrust in medico-pharmaceutical institutions. Exact question wordings and recoding can be found in Supplementary Materials 1.

**Alternative media consumption.** People who consume alternative forms of news may be more likely to be exposed to mis- or dis-information and less trustful of "mainstream" institutions [36,37]. Respondents were asked, "Of the following news-media outlets, which would you say is your primary source of news information?" Respondents who selected alternative news media outlets (*e.g.*, YouTube channels, Facebook) out of a list of possible sources were coded as relying on alternative news sources.

**Individualism scale.** We include a set of questions adapted from cultural cognition measures to capture respondents' level of individualism versus communitarianism, or their self-reported likelihood to act in an individualistic manner [38–40]. We created a dichotomous measure, with 1 representing respondents with highly individualistic values. Exact question wordings and recoding can be found in Supplementary Materials 1.

We anticipated that adding measures of trust, reliance on alternative media for news, and individualism would reduce the impact of race, ethnicity and vote choice on vaccine hesitancy.

## Experimental condition

We embedded an experimental treatment at the beginning of the survey, randomly exposing respondents to one of two news articles, modified from actual news stories. (see Supplement 5). A balance table demonstrating that there are no significant differences in treatment assignment across experimental arms on demographics, race, ethnicity, vote choice or income can be found in Supplement 2 (S2 Table in S2 File).

The two articles differed on three key dimensions: (1) whether racial disparities in COVID-19 impact were mentioned/emphasized including acknowledgments that "racism that is the root cause of this problem," (2) who an expert panel emphasized as recommended to be in the first priority group for vaccination and outreach, and (3) the race of the patient receiving a vaccine in the accompanying picture.

**Experimental 'racial prioritization' arm.** The experimental arm reported the conclusions of an expert panel that health care workers and other first responders should receive priority for the vaccine, while emphasizing (1) racial disparities in the impact of COVID-19 that are caused by racism and, accordingly, (2) emphasizing the panel's conclusion that disadvantaged minority groups should be a major focus of vaccine outreach. The photograph accompanying the article featured an female African-American receiving the vaccine from a white female health worker. This was the picture that accompanied the original news article that the story was adapted from.

**Control 'health worker priority' arm.** The control article used the same language about health care workers and first responders—but did *not* mention either COVID-19 disparities or prioritization based on race for vaccination outreach. Instead, the article emphasized recommendations for health care workers and essential workers to be in the first priority groups. The picture featured a white male receiving the vaccine from a white female health worker. This picture was also taken from the accompanying article that the story was adapted from. We considered that this condition could also increase intentions to vaccinate compared with no message at all, but we did not include a "pure" control group that did not read any prompt due to sample size and power considerations.

## Analysis

### Quantitative analysis

**Analysis of the survey experiment.** We examine whether respondents who received the experimental 'racial prioritization' news story reported differing vaccine intentions compared with respondents in the control 'health worker prioritization' condition. We also examine two dimensions of treatment heterogeneity: the interaction of race and ethnicity with the experimental prime, and the interaction of presidential vote choice in 2020 with the experimental prime. We estimated predicted probabilities to examine the interactions of the experimental arms with party identification and race-ethnicity to provide more interpretable results. We hypothesized that non-Hispanic Blacks receiving the experimental prime would express *higher intent to* vaccinate compared with non-Hispanic Blacks in the control group. We were unclear about effects among Hispanics. We also hypothesized that Trump voters receiving the experimental prime would express *lower* intent to vaccinate than Trump voters in the control group.

**Observational analyses of vaccine intentions.** To examine differences in vaccine hesitancy by race-ethnicity, vote choice, and other potential explanatory variables (separately from the effect of the experimental condition), we employed a stepwise regression approach with the experiment entered as a control variable to separate out effects of other variables in the model. Our main outcomes of interest are the odds ratios on hesitancy for non-Hispanic Blacks and Hispanics compared with non-Hispanic Whites, and 2020 Trump voters and non-voters/other voter, compared to 2020 Biden voters.

Model 1 analyzed the association between race/ethnicity and vote choice and vaccine hesitancy, with only the experimental condition as a control. In Model 2 we add demographic controls (sex, age, education and household income). In Model 3 we add other potential confounders (personal experience with COVID-19, co-morbidities, religiosity). In Model 4 we add the measures that may serve as "mediators" of the relationship between race, ethnicity, vote choice and vaccine intentions. In other words, these variables are likely to "explain" why we see relationships between these ascriptive or identity-based characteristics of individuals and their intentions to vaccinate (institutional trust scale, individualism, and alternative media consumption). In addition to examining the effects of these variables independently, we examine whether they reduce the magnitude and significance of race, ethnicity and/or vote choice on vaccination intentions. We expect their addition to the model should diminish or "soak up" any associations we observe between race, ethnicity or vote choice and intent to vaccinate.

**Robustness checks.** In addition to our primary analyses using ordered logit models, we also ran all models using logistic regression with a dichotomous dependent variable equaling one if the respondent said he/she definitely would not get the COVID-19 vaccine as soon as possible. We also ran all models with party identification (Democrat, Republican, Independent) rather than 2020 presidential vote choice. Results of these analyses are presented in Supplementary Materials 3 but were not qualitatively different. All analyses were conducted in Stata version 15.

### Qualitative analysis

These qualitative data are 'thin;' respondents typically wrote up to a sentence in response to the question "briefly describe why you would get the vaccine as soon as possible." This expands our understanding of vaccine intentions in the pre-rollout period but is still constrained by the brevity of the responses and inability to engage in back-and-forth dialogue.

Given the thinness of the data, all qualitative analyses were conducted in Excel. We conducted first-round coding inductively, retaining verbatim terms where possible. Each response

could be assigned more than one code if the respondent expressed multiple ideas in their response. First-round coding was further refined and aggregated, yielding 65 inductive codes. This resulted in six parent codes and forty-four child codes. All authors reviewed and commented on these codes and categorization and suggested revisions, which were then incorporated into a codebook (see Supplementary Materials 4 for codebook and more detail on the coding process).

## Results

### Quantitative results

**Descriptive results.** In this sample, non-Hispanic Blacks were 6 percentage-points more likely than non-Hispanic Whites or Hispanics to say that they definitely *would not* get a COVID-19 as soon as it is available to them: 18% of non-Hispanic Blacks said they definitely *would not* vaccinate as soon as possible compared with 12% for both non-Hispanic Whites and Hispanics (Fig 1 & Table 1). Non-Hispanic Whites were more likely to say that they definitely *would* vaccinate (52%) compared with non-Hispanic Blacks (33%) and Hispanics (42%) (Fig 1 & Table 1). Respondents who reported voting for Mr. Trump in 2020 were 8 percentage-points more likely to say they definitely *would not* vaccinate as soon as possible than 2020 Biden voters (19% versus 11%); Trump voters were similar to respondents who did not vote or voted for another candidate (18%) (Fig 2 & Table 1).

**Survey experiment.** We found no effect of the survey experiment in adjusted models on intent-to-vaccinate across the sample as a whole (see Table 2), and no significant effects at the 95% confidence level within racial, ethnic and voter subgroups (Fig 3). We consider this to be a true null.

**Stepwise regression.** Table 2 summarizes the results of the stepwise regression. Non-Hispanic Blacks were nearly twice as likely to be unlikely to vaccinate as soon as possible compared with non-Hispanic Whites (Model 3: OR = 1.82, p<0.01) adjusting for controls and

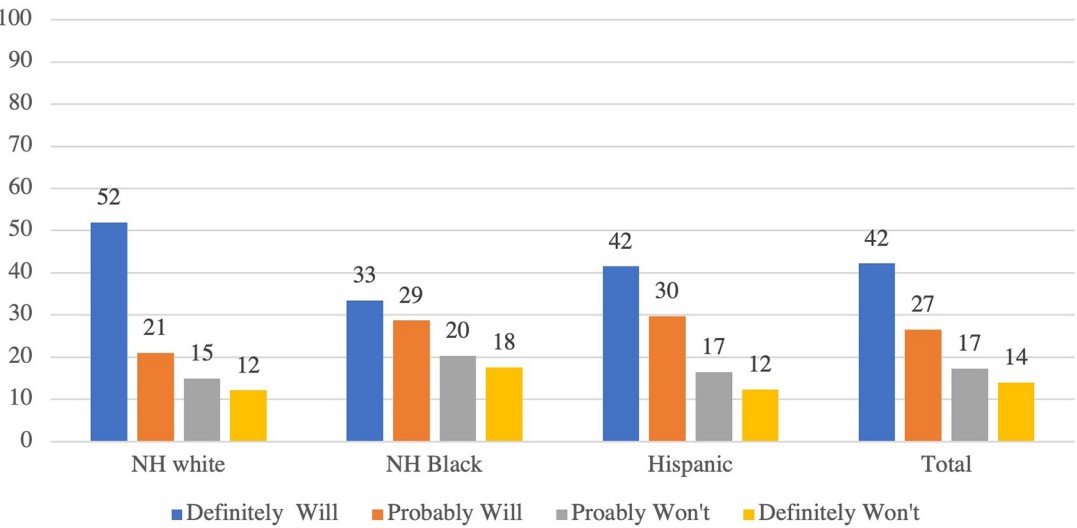

**Fig 1. COVID-19 vaccination intentions by race/ethnicity.**

**Table 1. Bivariate associations: Vaccine intentions and predictors.**

| | Definitely will get the Vaccine (N, Row %) | Probably Will Get the Vaccine (N, Row %) | Probably Won't Get the Vaccine | Definitely Not Get Vaccine (N, Row %) | Total (N, Row %) | Chi2, p-value |
|---|---|---|---|---|---|---|
| SURVEY EXPERIMENT | | | | | | |
| Broader Risk Group | 290 | 169 | 119 | 95 | 673 | |
| | 43.09 | 25.11 | 17.68 | 14.12 | 100 | |
| Racial Disparities | 282 | 190 | 114 | 94 | 680 | |
| | 41.47 | 27.94 | 16.76 | 13.82 | 100 | 0.702 |
| RACE/ETHNICITY | | | | | | |
| NH White | 238 | 99 | 69 | 56 | 462 | |
| | 51.52 | 21.43 | 14.94 | 12.12 | 100 | |
| NH Black | 148 | 127 | 90 | 78 | 443 | |
| | 33.41 | 28.67 | 20.32 | 17.61 | 100 | |
| Hispanic | 186 | 133 | 74 | 55 | 448 | |
| | 41.52 | 29.69 | 16.52 | 12.28 | 100 | <0.001 |
| VOTE CHOICE 2020 | | | | | | |
| Vote Biden | 366 | 226 | 139 | 94 | 825 | |
| | 44.36 | 27.39 | 16.85 | 11.39 | 100 | |
| Vote Trump | 86 | 42 | 30 | 36 | 194 | |
| | 44.33 | 21.65 | 15.46 | 18.56 | 100 | |
| Didn't vote/vote other | 120 | 91 | 64 | 59 | 334 | |
| | 35.93 | 27.25 | 19.16 | 17.66 | 100 | <0.001 |
| GENDER | | | | | | |
| Male | 336 | 178 | 96 | 57 | 667 | |
| | 50.37 | 26.69 | 14.39 | 8.55 | 100 | |
| Female | 236 | 181 | 137 | 132 | 686 | |
| | 34.4 | 26.38 | 19.97 | 19.24 | 100 | <0.001 |
| EDUCATION | | | | | | |
| High school | 155 | 107 | 64 | 61 | 387 | |
| | 40.05 | 27.65 | 16.54 | 15.76 | 100 | |
| Some college | 119 | 93 | 86 | 66 | 364 | |
| | 32.69 | 25.55 | 23.63 | 18.13 | 100 | |
| College | 139 | 95 | 60 | 40 | 334 | |
| | 41.62 | 28.44 | 17.96 | 11.98 | 100 | |
| Graduate degree+ | 156 | 64 | 23 | 22 | 265 | |
| | 58.87 | 24.15 | 8.68 | 8.3 | 100 | <0.001 |
| INCOME | | | | | | |
| <$50,000 | 202 | 161 | 122 | 116 | 601 | |
| | 33.61 | 26.79 | 20.3 | 19.3 | 100 | |
| $50,000-$100,000 | 106 | 65 | 47 | 31 | 249 | |
| | 42.57 | 26.1 | 18.88 | 12.45 | 100 | |
| >$100,000 | 264 | 133 | 64 | 42 | 503 | |
| | 52.49 | 26.44 | 12.72 | 8.35 | 100 | <0.001 |
| AGE | | | | | | |
| 18–29 | 155 | 129 | 79 | 55 | 418 | |
| | 37.08 | 30.86 | 18.9 | 13.16 | 100 | |
| 30–39 | 170 | 80 | 51 | 40 | 341 | |
| | 49.85 | 23.46 | 14.96 | 11.73 | 100 | |
| 40–49 | 95 | 52 | 26 | 31 | 204 | |

*(Continued)*

**Table 1.** (Continued)

| | Definitely will get the Vaccine (N, Row %) | Probably Will Get the Vaccine (N, Row %) | Probably Won't Get the Vaccine | Definitely Not Get Vaccine (N, Row %) | Total (N, Row %) | Chi2, p-value |
|---|---|---|---|---|---|---|
| | 46.57 | 25.49 | 12.75 | 15.2 | 100 | |
| 50–59 | 61 | 37 | 39 | 36 | 173 | |
| | 35.26 | 21.39 | 22.54 | 20.81 | 100 | |
| >60 | 91 | 61 | 38 | 27 | 217 | |
| | 41.94 | 28.11 | 17.51 | 12.44 | 100 | 0.002 |
| # OF COMORBIDITIES | | | | | | |
| 0 | 222 | 150 | 115 | 93 | 580 | |
| | 38.28 | 25.86 | 19.83 | 16.03 | 100 | |
| 1 | 216 | 124 | 75 | 67 | 482 | |
| | 44.81 | 25.73 | 15.56 | 13.9 | 100 | |
| 2 | 75 | 60 | 26 | 19 | 180 | |
| | 41.67 | 33.33 | 14.44 | 10.56 | 100 | |
| 3+ | 59 | 25 | 17 | 10 | 111 | |
| | 53.15 | 22.52 | 15.32 | 9.01 | 100 | 0.022 |
| IMMEDIATE FAMILY MEMBER DIE FROM COVID | | | | | | |
| No | 429 | 298 | 217 | 173 | 1,117 | |
| | 38.41 | 26.68 | 19.43 | 15.49 | 100 | |
| | 142 | 61 | 16 | 15 | 234 | |
| Yes | 60.68 | 26.07 | 6.84 | 6.41 | 100 | <0.001 |
| IMPORTANCE OF RELIGION | | | | | | |
| Relig Somewhat or not Important | 268 | 225 | 164 | 117 | 774 | |
| | 34.63 | 29.07 | 21.19 | 15.12 | 100 | |
| Religion Very Important | 304 | 134 | 69 | 72 | 579 | |
| | 52.5 | 23.14 | 11.92 | 12.44 | 100 | <0.001 |
| ALTERNATIVE MEDIA CONSUMPTION | | | | | | |
| Traditional Media | 496 | 299 | 181 | 131 | 1,107 | |
| | 44.81 | 27.01 | 16.35 | 11.83 | 100 | |
| Alternative Media | 76 | 59 | 52 | 58 | 245 | |
| | 31.02 | 24.08 | 21.22 | 23.67 | 100 | <0.001 |
| CONFIDENCE IN MEDICO-PHARMACEUTICAL INSTITUTIONS | | | | | | |
| No or very little confidence | 49 | 27 | 18 | 37 | 131 | |
| | 37.4 | 20.61 | 13.74 | 28.24 | 100 | |
| Some confidence | 123 | 125 | 96 | 67 | 411 | |
| | 29.93 | 30.41 | 23.36 | 16.3 | 100 | |
| A great deal of confidence | 202 | 148 | 89 | 58 | 497 | |
| | 40.64 | 29.78 | 17.91 | 11.67 | 100 | |
| Complete confidence | 198 | 59 | 30 | 27 | 314 | |
| | 63.06 | 18.79 | 9.55 | 8.6 | 100 | <0.001 |
| INDIVIDUALISM SCALE | | | | | | |
| Low 1 | 36 | 16 | 6 | 4 | 62 | |
| | 58.06 | 25.81 | 9.68 | 6.45 | 100.00 | |
| 2 | 343 | 209 | 127 | 94 | 773 | |
| | 44.37 | 27.04 | 16.43 | 12.16 | 100.00 | |

*(Continued)*

**Table 1.** (Continued)

|  | Definitely will get the Vaccine (N, Row %) | Probably Will Get the Vaccine (N, Row %) | Probably Won't Get the Vaccine | Definitely Not Get Vaccine (N, Row %) | Total (N, Row %) | Chi2, p-value |
|---|---|---|---|---|---|---|
| 3 | 180 | 126 | 90 | 70 | 466 |  |
|  | 38.63 | 27.04 | 19.31 | 15.02 | 100.00 |  |
| High 4 | 13 | 8 | 10 | 21 | 52 |  |
|  | 25.00 | 15.38 | 19.23 | 40.38 | 100.00 | <0.001 |

confounders, and remained more hesitant after introducing explanatory variables (institutional trust, alternative media, individualism) (Model 4: 2.01, p<0.01). Hispanics were also significantly more likely to be hesitant compared with non-Hispanic Whites though to a lesser degree than non-Hispanic Blacks (Model 3: OR = 1.37, p<0.05), and remained more hesitant after introducing controls and explanatory variables (Model 4: OR = 1.43, p<0.01).

Voting for Trump in 2020 was also associated with lower intention to vaccinate adjusting for controls and confounders (Table 2, Model 3: OR = 1.74, p<0.01). Not voting in 2020 or voting for other candidates besides Biden or Trump was also associated with a lower intent to vaccinate (Table 2, Model 3: 1.50, OR = p<0.05) However, this effect was attenuated and no longer significant after the introduction of mediating variables (Table 2, Model 4, OR = 1.15, 1.22 respectively, p>0.1).

In all specifications, gender was a strong predictor of hesitancy. Women were 95% more likely to be hesitant than men even after adjusting for conrols, confounders and potential mediators (Table 2, Model 4, OR = 0.95, p<0.01). Respondents older than age 40 were also more likely to be hesitant, including in models with explanatory mechanism variables included (Table 2, Model 4). Factors associated with reduced hesitancy included higher income, having 3 or more comorbidities, and having a family member die of COVID-19. These remained significant with the introduction of mediating variables. Watching alternative media was strongly associated with hesitancy (Model 4: OR = 1.93, p<0.01), as was distrust in medico-pharmaceutical institutions. Respondents who reported having very little trust in the medico-

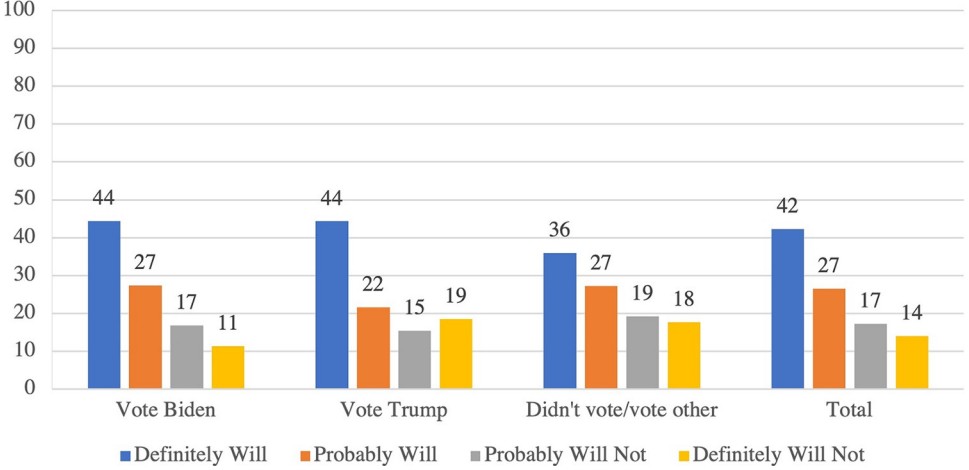

**Fig 2. COVID-19 vaccination intentions by vote choice 2020.**

**Table 2. Stepwise ordered logistic regression, disparities in vaccine hesitancy.**

| VARIABLES | No controls<br>odds ratio<br>Model 1 | Demographic Controls<br>odds ratio<br>Model 2 | Controls and Confounders<br>odds ratio<br>Model 3 | Controls, Confounders, and Mediators<br>odds ratio<br>Model 4 |
|---|---|---|---|---|
| experimental treatment | 1.04 | 1.03 | 1.03 | 0.98 |
| | (0.857–1.268) | (0.846–1.262) | (0.845–1.264) | (0.800–1.206) |
| **Race & Ethnicity** | | | | |
| NH White (ref) | ref | ref | ref | Ref |
| NH Black | 1.99*** | 1.79*** | 1.82*** | 2.01*** |
| | (1.545–2.551) | (1.376–2.327) | (1.396–2.368) | (1.524–2.645) |
| Hispanic | 1.32** | 1.24 | 1.37** | 1.43*** |
| | (1.035–1.691) | (0.952–1.603) | (1.049–1.778) | (1.094–1.881) |
| **Vote Choice 2020** | | | | |
| Vote Biden 2020 | ref | ref | ref | Ref |
| Vote Trump 2020 | 1.43** | 1.80*** | 1.74*** | 1.15 |
| | (1.054–1.933) | (1.314–2.459) | (1.271–2.390) | (0.827–1.612) |
| Did not vote 2020 | 1.45*** | 1.41*** | 1.50*** | 1.22 |
| | (1.146–1.827) | (1.103–1.795) | (1.170–1.913) | (0.947–1.573) |
| **Female** | | 1.83*** | 1.76*** | 1.95*** |
| | | (1.486–2.258) | (1.429–2.179) | (1.570–2.417) |
| **Age** | | | | |
| 18–30 (ref) | | ref | ref | Ref |
| 30–39 | | 0.92 | 0.98 | 1.24 |
| | | (0.698–1.218) | (0.738–1.294) | (0.927–1.654) |
| 40–49 | | 1.17 | 1.22 | 1.61*** |
| | | (0.840–1.625) | (0.877–1.701) | (1.142–2.265) |
| 50–59 | | 1.60*** | 1.54** | 1.92*** |
| | | (1.141–2.242) | (1.090–2.164) | (1.360–2.722) |
| 60+ | | 1.18 | 1.22 | 1.79*** |
| | | (0.855–1.638) | (0.874–1.708) | (1.262–2.536) |
| **Education** | | | | |
| < College (ref) | | ref | ref | Ref |
| Some College | | 1.70*** | 1.70*** | 1.71*** |
| | | (1.294–2.242) | (1.287–2.241) | (1.294–2.269) |
| College | | 1.27 | 1.21 | 1.26 |
| | | (0.949–1.708) | (0.903–1.634) | (0.937–1.707) |
| Graduate Degree | | 0.77 | 0.76 | 0.83 |
| | | (0.546–1.078) | (0.539–1.070) | (0.584–1.176) |
| **Income** | | | | |
| <$50,000 (ref) | | ref | ref | Ref |
| $50,000-$100,000 | | 0.71** | 0.71** | 0.8 |
| | | (0.529–0.939) | (0.535–0.954) | (0.600–1.078) |
| >$100,000 | | 0.56*** | 0.58*** | 0.68*** |
| | | (0.437–0.729) | (0.450–0.754) | (0.524–0.885) |
| **Co-morbidities** | | | | |
| 0 (ref) | | | ref | Ref |
| 1 | | | 0.80* | 0.81* |
| | | | (0.635–1.016) | (0.634–1.024) |
| 2 | | | 0.79 | 0.78 |

(*Continued*)

**Table 2.** (Continued)

| VARIABLES | No controls<br>odds ratio<br>Model 1 | Demographic Controls<br>odds ratio<br>Model 2 | Controls and Confounders<br>odds ratio<br>Model 3 | Controls, Confounders, and Mediators<br>odds ratio<br>Model 4 |
|---|---|---|---|---|
| | | | (0.572–1.083) | (0.561–1.073) |
| 3+ | | | 0.61** | 0.56*** |
| | | | (0.409–0.920) | (0.370–0.852) |
| Family member died COVID-19 | | | 0.46*** | 0.49*** |
| | | | (0.339–0.618) | (0.357–0.665) |
| Religion Very Important | | | | 0.64*** |
| | | | | (0.509–0.794) |
| Alternative Media | | | | 1.54*** |
| | | | | (1.183–2.013) |
| Trust in Medico-Pharm Institutions | | | | |
| Complete (ref) | | | | ref |
| A great deal | | | | 2.06*** |
| | | | | (1.532–2.766) |
| Some | | | | 3.20*** |
| | | | | (2.334–4.391) |
| Very little | | | | 3.76*** |
| | | | | (2.442–5.789) |
| Individualism score (high) | | | | 1.42*** |
| | | | | (1.146–1.768) |
| Observations | 1,353 | 1,350 | 1,348 | 1,347 |

*** p<0.01

** p<0.05

* p<0.1.

pharmaceutical institutions were nearly four times more likely to not intend to vaccinate (Table 2, Model 4, OR = 3.76, p<0.01). Scoring high on the individualism scale was also associated with a lower intent to vaccinate (Table 2, Model 4: OR = 1.42, p<0.01).

## Robustness checks

The results of the robustness checks with logistic regression and party identification were largely consistent with results from ordered logit models with vote choice, although with several caveats. For more details, see Supplementary Materials 3.

## Qualitative results

Free-response qualitative data—even thin, short-response answers—sheds light on what is top-of-mind (and willing-to-share) for respondents as they score their intent-to-vaccinate as soon as possible (see Supplmenatary Materials 4, S4 1–9 Fig in S1 File).

Among those who expressed willingness to get vaccinated as soon as possible, four themes emerged most strongly. The first is <u>protection</u>: almost half of respondents noted belief in the protective value against Covid-19 for themselves and/or their families as motivating them to get the shot as soon as possible. For example, one person noted, "I work outside the home and need to protect myself to protect my family," while another noted wanted to take proactive "steps to protect myself." Many wrote in the exact phrase "to protect myself and my family."

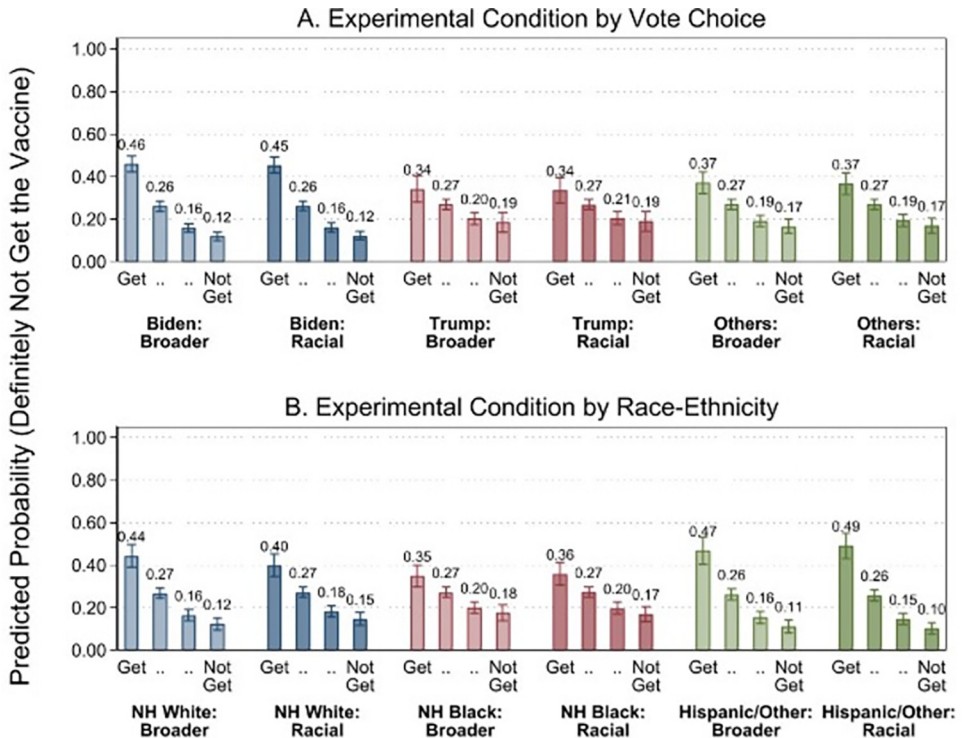

**Fig 3. Predictive margins, experimental conditions by race/ethnicity and vote choice.** Notes: DV = Definitely will not get the vaccine (3), Probably will not get the vaccine (2), Probably will get the vaccine (1), Definitely will get the vaccine (0). Controls included: gender, age, education, income, co-morbidities, family member died from COVID.

Second, and offering another dimension to why protection matters to them, some high intent-to-vaccinate respondents specifically noted high perceived susceptibility to Covid-19 or complicated sequelae from infection, judging themselves at high risk for negative outcomes given their current health status (*e.g.*, "I am black and old") and/or profession (*e.g.*, "I am an essential worker"). A few in this situation noted being "scared of Covid" and that vaccination would bring "piece of mind."

An additional set of responses (roughly a tenth in each case) evoked a sense of <u>collective responsibility</u>, framing vaccination as "doing [one's] part" or as a "public duty," extending the idea of protection beyond the self or household. Finally, people expressed a <u>desire to return to normal life</u>, and understood their vaccination decision to be a key route to "be free." In general, these themes were articulated with similar frequency across our demographic and political groups of interest, though non-Hispanic Whites chose to write about both about perceived susceptibility to Covid-19 and collective responsibility more than other groups (see S4 8 Fig in S1 File).

Among those who did not want to get vaccinated as soon as possible, two sometimes overlapping themes came out most strongly: <u>concerns about the safety</u> of the new Covid-19 vaccination and an explicit desire to "<u>wait and see</u>" about potential side effects of the vaccination. This included wanting others to "get it first" and to be "cautious" about their own decision. A subset wanted to hear about the safety of the vaccine specifically for people like them (age, race, disease status) and to hear it from their own or another healthcare professional. Others noted that it was "too soon" or "rushed;" a few among these cited a specific mental model that a vaccine "should take ten years to develop" and therefore needed "further testing." A small

(n = 8) subset of these—all non-Hispanic Black respondents—explicitly stated not wanting to "be a guinea pig" or a "lab rat."

Among othose who said they would *never* get the vaccine, a small number of respondents wrote both about generalized vaccine refusal or specific rumors circulating along with the Covid-19 pandemic. Generalized vaccine concerns and refusal—expressed by Biden-voters, Trump-voters, and non-voters—were articulated as not "liking," "trusting," "believing in," or being "keen on" vaccines across the board. Meanwhile, respondents (n = 12) who elaborated on never wanting to get the vaccine with conspiratorial references included skepticism about the vaccine (*e.g.*, "I have concerns about microchips in the vaccine") and about Covid-19 (*e.g.*, "I think there is something not kosher about this whole alleged 'crisis.'"). These respondents identified as Trump-voters. Unexpectedly, no one wrote in with specific concerns about the vaccine and pregnancy and/or fertility.

Though there were differences across race, ethnic and partisan groups in the themes mentioned, the differences were smaller than differences in the close-ended questions, suggesting that all race-ethnic groups share similar underlying motivations and concerns but may weigh these differently and/or express those differently on a Likert-type scale—a measurement point that warrants further investigation in future research.

## Discussion

A large majority (86%) of respondents in this diverse sample of New York-based adults expressed some degree of willingness to be vaccinated for COVID-19, although an important segment (14%) of the sample stated that they definitely would not get the vaccine as soon as possible. Consistent with previous literature [41–43], we found racial and ethnic differences in vaccine hesitancy in the pre-rollout period, with non-Hispanic Blacks being 6 percentage points more likely to say that they definitely would not vaccinate as soon as possible for COVID-19 compared with non-Hispanic Whites and Hispanics. Hispanics were also more likely to be hesitant than non-Hispanic Whites, with fewer saying they would definitely vaccinate as soon as possible (42% compared with 52%). Trump voters were 8 percentage points more likely to say they definitely would not vaccinate compared with Biden voters and those who as well as people who did not vote in 2020 or voted outside of the two main candidates were 7 percentage points more likely to say they definitely would not vaccinate. This is consistent with literature showing heightened hesitancy among people who identify as conservative and/or Republican [6,44–48].

These descriptive findings should be interpreted in light of the situation in November/Decmeber of 2020 when the survey was fielded. This was a period before COVID-19 vaccines became widely available and began being rolled out to essential workers, which started in January 2021. People's intentions were more hypothetical at this point since vaccines had not yet been made available yet. It was also a period just following the 2020 Presidential election when Donald Trump lost the election. Media reporting at this time suggested that the vaccine may have been rushed for political reasons to re-open the economy [49]. Political right opposition to vaccination may have therefore been lower at this time given its association with the Trump administration, but also changing given his political loss.

To examine whether a racial justice message about vaccination might reduce hesitancy in non-Hispanic Blacks, but increase hesitancy in non-Hipanic Whites, we experimentally tested the effect of reading a news article about racial prioritization for the COVID-19 vaccine on intent to vaccinate. These messages did not affect expressed willingness to vaccinate overall or for any race, ethnic or voting group; there was no positive effect among non-Hispanic Blacks/ Biden voters nor was there a backlash effect among non-Hispanic White/Trump voters. These

results run counter to our hypotheses that exposure to a message acknowledging justified mistrust in African-Americans could reduce hesitancy in non-Hispanic blacks and Biden voters/ Democrats, and would likely have no effect or could even possibly backfire among non-Hispanic Whites and Trump voters/Republicans. The overarching null findings should be interpreted in light of the fact that there was relatively limited power to detect small heterogeneous treatment effects in this sample. Furthermore, other studies have examined the moderating effects of racial attitudes and prior knowledge of COVID-19 disparities on similar exposures [31]. Heterogeneity in treatment effects according to these factors could account for the null effect in this setting.

In observational analyses of the survey data, we found that the lower intent to vaccinate among Trump voters and non-voters was no longer significant after controlling for a hypothesized set of mediators, including distrust in medico-pharmaceutical institutions, consumption of alternative media, and individualistic values. This suggests that hesitancy based on political partisanship can largely be explained by these factors. Distrust has been highlighted both as an independent predictor of support for former President Trump [50] and a predictor of hesitancy [51]. This pattern may highlight an underlying latent set of trust-related characteristics driving both attitudes. This group has been shown to have more anti-establishment attitudes [35] and may also be a target for anti-vaccine messages on social media [26].

Unlike vote choice, the significance of race and ethnicity persisted even after inclusion of measures of mistrust, alternative media use, and individualism. It is possible that the particular measures of mistrust were insufficient to capture the nuanced mechanisms through which distrust in these institutions shapes COVID-19 vaccine attitudes. Prior studies have found that mistrust among Black populations in the U.S. may be further exacerbated by the general sociopolitical climate in the US [52], or potential confounding with socioeconomic status that is diificult to measure accurately and adjust for in observational studies with online samples (*e.g.*, lack of financial resources or housing insecurity) [53].

While we found race, ethnicity, 2020 vote choice, and partisanship to be significant predictors of hesitancy, gender and age were even more consistently and strongly associated with hesitancy even after adjustment for other factors, including institutional mistrust. This is consistent with previous literature that has shown women to be more vaccine hesitant [45]. The age effect is likely representative of a cohort effect whereby there are generational differences in acceptance of COVID-19 vaccines, but this cannot be separated in observational studies. Other studies have found an inverted U-shaped relationship in the relationship between age and vaccine hesitancy as well [54]. This suggests that in addition to considering race and ethnicity, gender and certain age cohorts may be additional groups that can be targeted for additional vaccine promotion.

Using thin qualitative data of how respondents explain briefly their intent-to-vaccinate choice, we found hesitancy views were present in all race, ethnic and voter groups and that the major categories of objections did not manifest differently across groups. The complexity of vaccine hesitancy and its presence across racial, ethnic, gender, and age groups highlights the need for a deeper understanding of the relationship between the expression of hesitant attitudes and their translation into intention to vaccinate.

The qualitative responses also revealed relatively few explicit references to politics, although trust in the political system and the skepticism that a "safe" vaccine could be produced in such a short time frame was prevalent. This may be surprising given that the politics of vaccination has been salient throughout COVID-19 in the United States over the period preceding the survey, which took place several weeks after the US presidential election in November 2020. Despite widespread concern about the rise of misinformation and fake news spread through social media, and potential susceptibility of people already mistrustful of mainstream

institutions to mis/dis-information [55–58], write-in references to outright conspiracies were rare and were no more likely to be held by racial or ethnic minority groups. More frequent (although still rare) references to conspiracy theories among Trump supporters may reflect a sense of mistrust among this group, or greater exposure to misinformation.

The fact that ascriptive characteristics of individuals, i.e., characteristics corresponding to one's external identity assigned by society, remained significant and even increased in magnitude after adjusting for potential confounders and potential mediating factors, including mistrust, deserves further scrutiny. This suggests that these identity-based characteristics are meaningful in themselves, and cannot be reduced to a set of opinions or ideas often theorized to explain or mediate these effects (i.e., lack of trust). Furthermore, the fact that the experimental condition did not significantly alter vaccination intentions also suggests that message framing acknowledging justified mistrust in minorities may not be sufficient on its own to overcome hesitant attitudes. Future research may want to assess how much impact message framing has on its own compared with other factors known to be important in vaccine uptake including peer influence, communication from physicians and mandates.

## Limitations

The sample was not designed to be representative of the population of New York State. The web-based sampling strategy and the goal of oversampling underrepresented racial and ethnic groups allowed for robust analysis of intra-group patterns, but cannot be easily extrapolated to the public at large. The qualitative responses were based on a large sample of qualitative respondents (<1000) but from a relatively short set of open-ended, short-answer qualitative responses from this group. The survey experiment may have been underpowered to detect modestly-sized effects especially for heterogeneous treatment effects.

Another significant limitation is that the image that was included that accompanied the newspaper article highlighting justified mistrust of the medical establishment showed a white doctor inoculating a black patient. This racial discordance between doctor and patient could have weakened the message aimed at easing mistrust as it may have caused respondents to consider ongoing medical power differentials related to concerns about medical racism. Future studies that may wish replicate this experiment or related experiments could consider similar designs using racially concordant doctors and patients to mitigate this possible source of bias. Pre-testing of the image could have helped to assess how the racial discordance plays into the overall message. However, a recent paper by Gadarian et al. shows that same-race/ethnicity expert endorsements had no effect on nonwhite or white respondents' willingness to get the COVID-19 vaccine, to encourage others to get the vaccine, or to learn more or share information with others [11]. This suggests that racial-concordance between the doctor and the patient may not have influenced the null finding. Future research could also use larger samples to examine how different messages addressing justified mistrust and acknowledging structural racism can influence attitudes towards vaccination and intention-to-vaccinate, or can lead to backfiring.

A final limitation is that we did not include a "true" control condition that excludes any exposure to a news article about prioritization given our limited budget. We therefore cannot say for certain whether respondents' intent to vaccinate would have looked different if they had not seen any article at the outset of the survey. For the purpose of this paper, we have treated the condition about prioritization of health workers and other priority groups (but without any reference to race or ethnicity) as the control. While the overall message conveyed in both is quite similar as both arms cite the recommendations of the U.S. Vaccine Advisory Panel, there are several differences in the wording and listing of groups being prioritized

between the two conditions. To view the full set of ways that the treatment and control arms differ in wording, the full treatment and control arms are included in the Supplementary Materials. However, the title of the articles is identical and is focused on who should get priority for the vaccines when initial supply was limited with the primary difference being that the experimental condition discusses prioritization of low-income minorities and acknowledges racism is the root cause of health disparities. Both articles are adapted from actual news stories and therefore are realistic exposures that the public might consume.

## Conclusions

Generalized vaccine hesitancy has been observed in many groups and geographies. In the context of COVID-19, while initial concerns about vaccine hesitancy focused on non-Hispanic Blacks, increasingly concern has fixated on the low vaccination rates among Republicans and supporters of former President Trump. This polarization of attitudes poses challenges for public health strategies to build vaccine confidence. To increase vaccine confidence, health communication scholars recommend acknowledging disparities and the reality that medical mistrust has a basis in histories of discrimination. However, it is possible that this framing could come at the expense of reducing hesitancy in other groups.

We find that an experimental message conveying racial vaccine prioritization and acknowledging structural racism in COVID-19 disparities has no effect on intent to vaccinate overall or in subgroups. Moreover, we find lower intentions to vaccinate among non-Hispanic Blacks, Hispanics, females and certain age cohorts, that are not accounted for by higher degrees of institutional mistrust, whereas for Trump voters and non-voters, the correlation between this identity and hesitancy is no longer evident after controlling for low trust and ideological factors.

These results reinforce the idea that for some groups, strong hesitancy views may be relatively fixed and difficult to change at least with relatively simple messaging campaigns. However, qualitatively we also find strong similarities in the kinds of reasons offered by individuals across racial and ethnic groups, suggesting that for some "softer" elements of hesitancy (notably the "wait and see" group), including that general observation of safe and effective vaccination among respondents' social circles, may reduce hesitancy. When introducing new technologies in the future, health communicators might consider how to make the (lack of) side effects and limited disruption to daily life more visible. More positively, the racial-prioritization prime did not backfire and produce lower intentions in non-Hispanic Whites nor Trump voters, suggesting that framing minorities as disproportionately at risk and deserving of prioritization is not necessarily a polarizing or zero-sum message.

In the short run, people may eventually choose to vaccinate in spite of hesitancy in order to get back to "normal" life, but the deep and enduring mistrust of public institutions that underlie hesitancy, are unlikely to reduce any time soon and may to continue to hinder public health efforts into the future. More attention should be given to how to build trust in medical institutions and to depoliticize and depolarize opinions on vaccines, so that this and future vaccination campaigns will be starting from a period of high trust among all social, ethnic and racial groups.

## Supporting information

**S1 File.**
(ZIP)

**S2 File.**
(DOCX)

**S3 File.**
(DOCX)

## Author Contributions

**Conceptualization:** Ashley Fox, Yongjin Choi, Heather Lanthorn, Kevin Croke.

**Data curation:** Ashley Fox, Yongjin Choi, Heather Lanthorn, Kevin Croke.

**Formal analysis:** Ashley Fox, Yongjin Choi, Heather Lanthorn, Kevin Croke.

**Funding acquisition:** Ashley Fox, Yongjin Choi, Kevin Croke.

**Investigation:** Ashley Fox, Yongjin Choi, Kevin Croke.

**Methodology:** Ashley Fox, Yongjin Choi, Heather Lanthorn, Kevin Croke.

**Project administration:** Ashley Fox, Yongjin Choi, Heather Lanthorn.

**Resources:** Ashley Fox.

**Software:** Ashley Fox, Yongjin Choi.

**Supervision:** Ashley Fox, Kevin Croke.

**Validation:** Ashley Fox, Heather Lanthorn, Kevin Croke.

**Visualization:** Ashley Fox, Yongjin Choi.

**Writing – original draft:** Ashley Fox, Yongjin Choi, Heather Lanthorn, Kevin Croke.

**Writing – review & editing:** Ashley Fox, Yongjin Choi, Heather Lanthorn, Kevin Croke.

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
