## [Decision Letter · Decision Letter 0]

10 Feb 2022

PONE-D-21-32376Racial, ethnic and partisan differences in COVID-19 vaccine hesitancy, its sources and potential solutions in a diverse samplePLOS ONE

Dear Dr. Fox,

Thank you for submitting your manuscript to PLOS ONE. After careful consideration, we feel that it has merit but does not fully meet PLOS ONE’s publication criteria as it currently stands. Therefore, we invite you to submit a revised version of the manuscript that addresses the points raised during the review process.

ACADEMIC EDITOR:Hello. First I want to express that it is frustrating to have to wait so long for a response for review. As someone who has been in your position, I apologize for the wait. I invited over 20 reviewers with most of them rejecting to review. Fortunately, we were able to secure the minimum of two reviewers. The first reviewer recommended a rejection and the other recommended a major revision. Upon reading the first reviewer's report, most of the comments could be regarded as major revisions, though some of them may require revisions that go beyond the scope of what is a typical major revision, thus perhaps leading them to decide on a rejection.  As someone who has also published on vaccine hesitancy, I think your study has content that is worth releasing to the public. I invite you and your team to attempt a major revision based on the comments below.

We look forward to receiving your revised manuscript.

Kind regards,

Andrew G Wu

Academic Editor

PLOS ONE

Journal Requirements:

This research was supported by funding from the Community Engaged COVID-19 Health Disparities Researcher group at UAlbany

AMF received funding for the data collection of this project from the Community Engaged COVID-19 Health Disparities Researcher group at the University at Albany. This was an internal university grant made possible through funding from the State of New York.  

Reviewers' comments:

Reviewer's Responses to Questions

**Comments to the Author**

1. Is the manuscript technically sound, and do the data support the conclusions?

Reviewer #1: Partly

Reviewer #2: Yes

2. Has the statistical analysis been performed appropriately and rigorously? 

Reviewer #1: Yes

Reviewer #2: Yes

3. Have the authors made all data underlying the findings in their manuscript fully available?

Reviewer #1: Yes

Reviewer #2: Yes

4. Is the manuscript presented in an intelligible fashion and written in standard English?

Reviewer #1: Yes

Reviewer #2: No

5. Review Comments to the Author

Reviewer #1: This is an interesting study that examines vaccine hesitancy using a mixed methods approach combining survey analysis, qualitative data analysis of open-ended questions, and a survey experiment. In spite of the strengths of the methods and analysis, there are a number of problems with the manuscript that lead me not to recommend it for publication in its present form.

(1) The literature review does not sufficiently examine results of other published studies examining COVID-19 vaccine hesitancy in the US over the past year. Situating the present findings in light of more data from around the US as well as comparisons to different time periods of the pandemic would be very important to get a better sense of how to interpret the findings. E.g., the present study seems to support other findings about Republicans and African Americans expressing more hesitancy. But have other studies similarly found women to be more hesitant than men or older people more likely to be hesitant, or are the findings elsewhere different? See for a review of some of this literature: Wang, Ying, and Yu Liu. "Multilevel determinants of COVID-19 vaccination hesitancy in the United States: a rapid systematic review." Preventive Medicine Reports (2021): 101673.

(2) The study suffers from serious limitations regarding generalizability, due to the time period of the data collection as well as sampling methodology. Neither the abstract nor the title notes that the study is focused on a non-representative sample of New York State residents during November-December 2020, which is before the COVID-19 vaccine was approved. There are serious limitations with what we can learn about vaccine hesitancy from this study.

First, even though NYS is demographically and politically diverse, there’s no weighting applied in this dataset that could allow the sample to be generalized to the state population. Oversampling blacks and Hispanics is great, but again without sample weights, the results of a similar analysis may look very different in a weighted sample. The paper needs to also say something about how the results of a similar study conducted elsewhere in the US might look.

Second, the time period is an important issue. After the approval of the vaccine, vaccination attitudes changed considerably over the span of a few months as more information (and misinformation) about mRNA vaccines started to circulate. mRNA vaccines have introduced new kinds of fears and hesitancies which are crucial for us to understand for the very reasons articulated in the manuscript, but the study is unable to offer such insights. It is also the case that since early 2021, African American communities have taken leadership in driving vaccination efforts, even using churches as vaccination centers. So would the same study, if carried out four or six months later, have yielded different results? This is an important consideration that the authors need to address.

This is not to say the study cannot make important contributions; they are just narrower in scope and need to be stated as such.

Minor points:

The coding of the qualitative data needs to distinguish between distrust or refusal of vaccines in general vs. distrust of this specific vaccine (e.g., “I don’t believe in getting that vaccine”, “I do not trust the vaccine”). Those seem to be conflated in the current study. It is also difficult to tell how certain statements suggesting conspiracy theories (e.g., concerns about microchips) do not also underlie some of the vaguer statements of distrust (i.e., there may be more conspiracy theorists who just didn’t articulate their reasons). Also given the non-representativeness of the sample, I’m not sure what to make of the distributions in the qualitative data; just because a particular theme stands out in this dataset, that doesn’t mean the opinion it reflects is proportionately prevalent in the broader population.

Regarding the experiment, I found it striking (and this is not mentioned in the manuscript at all) that in the image of the African American person receiving the vaccine, the person administering the vaccine is white. If part of the reason for vaccine hesitancy is ongoing racism experienced by blacks in the healthcare system, then this prime would only reinforce such mistrust. Might the results have been different if the physician in the image had been black? Did the researchers conduct any cognitive testing of these images in order to make sure they did not have unintended effects/interpretations?

Reviewer #2: Thank you for the opportunity to review this article. 

I believe the topic is important and that differential uptake in COVID-19 vaccination will exacerbate pre-existing socioeconomic inequalities amongst marginalised ethnic and social groups. This study adds to our understanding or confirms our knowledge to an extent. Many thousands of unvaccinated people have died in the USA and elsewhere despite the availability of safe, effective vaccines that can significantly reduce the risk of serious disease and death. The topic is therefore important, timely and helpful. 

General feedback: the article lacks clarity in places. The authors also assume that the readers will not exactly what they are talking about but of course, this should not be assumed and the language should be made accessible. The authors just mention terms like 'vaccine hesitancy' without even an attempt to explain what it means. It should be defined and an appropriate citation is given. Acronyms are widely used without expanding what they stand for in the main body of the article. For example, in the introduction, it says NH Black. I understand that the authors have explained in the abstract but this should be done in the introduction as well. Non-Hispanic is one word but it will add to the clarity. I don't think ASAP is an appropriate acronym in an academic paper and should not be used. 

I have the following specific feedback for the authors. 

Abstract:It is not clear what the authors mean by 'the experimental manipulation of a vaccine promotion message'. This needs to be written more clearly in the abstract without having to look at the methodology and results for what they mean. Where the authors mention 'ascriptive' they need to open a bracket and list some of those characteristics. For example ascriptive characteristic (i.e., race, gender....).  

Ethics:I am not sure why the institutional review board is anonymised. For a survey of this nature, I believe we more information about ethics and the approval reference. 

Methods:

The regression analyses in controlling for potential confounding factors and adjusting for demographic characteristics are good and improve the validity and reliability of the results. 

Results:

The odds ratio in NH blacks and to a certain extent Hispanics and NH Whites Trump voters are quite high and significant. The statistical analyses look fine to me but will need to be double-checked by a statistician. 

There is a very high level of vaccine hesitancy in women in this sample. Were these reproductive-aged women? Low intent to vaccinate is particularly high in pregnant or reproductive-aged women as multiple studies have shown. It would be important to comment on that given how serious COVID-19 infection is in pregnancy. 

Free text responses - the authors rightly acknowledge that they could not do proper qualitative analyses on these. I believe it is not right to impose a quantitative framework on a qualitative response. It does not matter what percentage of people mentioned lack of trust it is the depth, richness, and complexity are would be uncovered through qualitative responses. 

Discussion:

The limitations section is adequate. 

The authors say that higher vaccine hesitancy among Non-Hispanic Blacks is not entirely accounted for institutional mistrust compared to NH Whites Trump votes. However, it would be important to discuss from existing literature other possible reasons for this.

6. PLOS authors have the option to publish the peer review history of their article (what does this mean?). If published, this will include your full peer review and any attached files.

Reviewer #1: No

Reviewer #2: No

---

## [Author Response · Author response to Decision Letter 0]

16 May 2022

May 2, 2022

Dear Reviewers,

Thank you for providing us with these helpful comments on our paper, which have allowed us to substantially strengthen and update the manuscript. Some of the issues that you raised about the literature review being out of date and the overall contribution have, at least in part, to do with the lengthy peer-review process that the article has gone through, which have made the findings less timely than when we first undertook the research. Nonetheless, we do believe that the research community is better off having access to our research and its approach than not- for instance to be able to replicate the results or try similar experiments in different samples. Throughout we have tried to update the manuscript, better clarify what we see as the main contributions of the project and be more forthright about some of its limitations. We have substantially revised the manuscript throughout. 

Below we outline our response to your comments in red font.

Sincerely, 

The Authors

Reviewers' comments:

5. Review Comments to the Author

Reviewer #1: This is an interesting study that examines vaccine hesitancy using a mixed methods approach combining survey analysis, qualitative data analysis of open-ended questions, and a survey experiment. In spite of the strengths of the methods and analysis, there are a number of problems with the manuscript that lead me not to recommend it for publication in its present form.

(1) The literature review does not sufficiently examine results of other published studies examining COVID-19 vaccine hesitancy in the US over the past year. Situating the present findings in light of more data from around the US as well as comparisons to different time periods of the pandemic would be very important to get a better sense of how to interpret the findings. E.g., the present study seems to support other findings about Republicans and African Americans expressing more hesitancy. But have other studies similarly found women to be more hesitant than men or older people more likely to be hesitant, or are the findings elsewhere different? See for a review of some of this literature: Wang, Ying, and Yu Liu. "Multilevel determinants of COVID-19 vaccination hesitancy in the United States: a rapid systematic review." Preventive Medicine Reports (2021): 101673.

Thank you for noting this. We agree that our literature review section needs to be updated to reflect current knowledge of the COVID-19 vaccine landscape in the US. We have tried to update and synthesize throughout the introduction including incorporating the helpful review article you cite (Wang, Ying, and Liu, 2021). We have substantially updated the literature throughout and tried to better situate the findings in the time period in which the survey/survey experiment was undertaken (Nov/Dec 2020 prior to vaccine roll-out and just following the 2020 Presidential elections).

(2) The study suffers from serious limitations regarding generalizability, due to the time period of the data collection as well as sampling methodology. Neither the abstract nor the title notes that the study is focused on a non-representative sample of New York State residents during November-December 2020, which is before the COVID-19 vaccine was approved. There are serious limitations with what we can learn about vaccine hesitancy from this study.

We do not dispute this point about the limitations of what we can learn from this study, including that it is from a non-representative sample of New York State residents and at a single time-point in the pandemic before vaccines were available to everyone. However, we would like to clarify that the study was undertaken primarily as a survey experiment and to collect short qualitative information on reasons that people give for desiring/hesitating to vaccinate, neither of which require representative samples to make broader inferences.

We have tried make these points about its contribution more salient throughout, including acknowledging this in the limitations section.

We have also revised the title to better signal what we view as the main contribution of this piece (i.e., the survey experiment and qualitative findings) and the specificity of the sample/time period. We have revised the title as follows:

“Does Framing COVID-19 in Terms of Disparities Reduce or Increase Vaccine Hesitancy? Evidence from a Survey Experiment in a Diverse Sample in New York State prior to vaccine roll-out”

Finally, we have reorganized our presentation of the order of our results, starting with the survey experiment followed by the descriptive survey analysis and the qualitative results. We hope these revisions can better contextualize the contribution with greater emphasis on the survey experiment.

First, even though NYS is demographically and politically diverse, there’s no weighting applied in this dataset that could allow the sample to be generalized to the state population. Oversampling blacks and Hispanics is great, but again without sample weights, the results of a similar analysis may look very different in a weighted sample. The paper needs to also say something about how the results of a similar study conducted elsewhere in the US might look.

Second, the time period is an important issue. After the approval of the vaccine, vaccination attitudes changed considerably over the span of a few months as more information (and misinformation) about mRNA vaccines started to circulate. mRNA vaccines have introduced new kinds of fears and hesitancies which are crucial for us to understand for the very reasons articulated in the manuscript, but the study is unable to offer such insights. It is also the case that since early 2021, African American communities have taken leadership in driving vaccination efforts, even using churches as vaccination centers. So would the same study, if carried out four or six months later, have yielded different results? This is an important consideration that the authors need to address.

This is not to say the study cannot make important contributions; they are just narrower in scope and need to be stated as such.

We have tried to tone done the contributions in various places, starting with the title, which we renamed “Does Highlighting COVID-19 Disparities Reduce or Increase Vaccine Intentions? Evidence from a Survey Experiment in a Diverse Sample in New York State Prior to Vaccine Roll-out.” We now discuss more about evolving trends in hesitancy and vaccine uptake based on updated literature review. Further, by emphasizing the survey experiment, we are contributing more broadly to emerging literature on whether messages emphasizing disparities are harmful or helpful to vaccine promotion. We believe that even though our experimental results on this front are null, it is important to report this. 

Minor points:

The coding of the qualitative data needs to distinguish between distrust or refusal of vaccines in general vs. distrust of this specific vaccine (e.g., “I don’t believe in getting that vaccine”, “I do not trust the vaccine”). Those seem to be conflated in the current study. It is also difficult to tell how certain statements suggesting conspiracy theories (e.g., concerns about microchips) do not also underlie some of the vaguer statements of distrust (i.e., there may be more conspiracy theorists who just didn’t articulate their reasons). Also given the non-representativeness of the sample, I’m not sure what to make of the distributions in the qualitative data; just because a particular theme stands out in this dataset, that doesn’t mean the opinion it reflects is proportionately prevalent in the broader population.

Thank you for this comment. We have made several adjustments to the paper to make our qualitative work clearer, including writing out the precise questions we posed in the methods and rewriting our section on the Qualitative results- see p. 24-26. We also took out the Figures and put those in the Supplementary materials for those who want more details on relative frequencies of statements.

Regarding the experiment, I found it striking (and this is not mentioned in the manuscript at all) that in the image of the African American person receiving the vaccine, the person administering the vaccine is white. If part of the reason for vaccine hesitancy is ongoing racism experienced by blacks in the healthcare system, then this prime would only reinforce such mistrust. Might the results have been different if the physician in the image had been black? Did the researchers conduct any cognitive testing of these images in order to make sure they did not have unintended effects/interpretations?

Yes, this was an oversight on our part. The image was the actual image associated with the newspaper article that we built the experiment from, but we could have been more thoughtful about whether the racial discordance in the image would work against our hypotheses. We agree that the experiment would have benefited from cognitive testing of these images, but we did not undertake this. We have now listed this as a limitation in the Limitations section and something that future research should consider if they try to replicate this experiment or related experiments.

See Limitations section, p. 31: “Future studies that may wish to replicate this experiment with a racially concordant doctor and patient or related experiments could consider similar designs using racially concordant doctors and patients to mitigate this possible source of bias. Pre-testing of the image could have helped to assess how the racial discordance plays into the overall message.”

Reviewer #2: Thank you for the opportunity to review this article. 

I believe the topic is important and that differential uptake in COVID-19 vaccination will exacerbate pre-existing socioeconomic inequalities amongst marginalised ethnic and social groups. This study adds to our understanding or confirms our knowledge to an extent. Many thousands of unvaccinated people have died in the USA and elsewhere despite the availability of safe, effective vaccines that can significantly reduce the risk of serious disease and death. The topic is therefore important, timely and helpful. 

General feedback: the article lacks clarity in places. The authors also assume that the readers will not exactly what they are talking about but of course, this should not be assumed and the language should be made accessible. The authors just mention terms like 'vaccine hesitancy' without even an attempt to explain what it means. It should be defined and an appropriate citation is given. Acronyms are widely used without expanding what they stand for in the main body of the article. For example, in the introduction, it says NH Black. I understand that the authors have explained in the abstract but this should be done in the introduction as well. Non-Hispanic is one word but it will add to the clarity. I don't think ASAP is an appropriate acronym in an academic paper and should not be used. 

Thanks for these points. It is true that we perhaps assumed a level of knowledge of vaccine hesitancy that may not be appropriate for a more general readership journal like PLoS One. We have added a definition of vaccine hesitancy to the introductory paragraph. We have also increased our own precision by clarifying that we are actually measuring intent to vaccinate against COVID-19. We have also spelled out the abbreviations the reviewer mentions (non-Hispanic and as soon as possible). In general, we have substantially re-written the article to the point where the track changes shown that a large portion of the manuscript has been completely rewritten and updated.

I have the following specific feedback for the authors. 

Abstract:It is not clear what the authors mean by 'the experimental manipulation of a vaccine promotion message'. This needs to be written more clearly in the abstract without having to look at the methodology and results for what they mean. Where the authors mention 'ascriptive' they need to open a bracket and list some of those characteristics. For example ascriptive characteristic (i.e., race, gender....). 

Thanks for these suggestions. We have amended the sentence in the abstract as follows: 

“Experimental manipulation of a racial equity vaccine promotion message via an online survey experiment had no effect on intention-to-vaccinate in the full sample or in racial, ethnic and partisan subsamples.”

We have also clarified the term “ascriptive characteristics” per the Reviewer’s suggestion in the Highlights and in other places where it is mentioned in the manuscript. Ascriptive characteristics refer to people’s “outward identities” as opposed to underlying values traits/personality characteristics. 

Ethics:I am not sure why the institutional review board is anonymised. For a survey of this nature, I believe we more information about ethics and the approval reference. 

This was originally done for anonymous peer review purposes, but we have now added more detailed institutional review board information in the text now:

p. 9: “This study was approved by the Institutional Review Board of the University at Albany (IRB study number, 20X258).” 

Methods:

The regression analyses in controlling for potential confounding factors and adjusting for demographic characteristics are good and improve the validity and reliability of the results. 

Results:

The odds ratio in NH blacks and to a certain extent Hispanics and NH Whites Trump voters are quite high and significant. The statistical analyses look fine to me but will need to be double-checked by a statistician. 

There is a very high level of vaccine hesitancy in women in this sample. Were these reproductive-aged women? Low intent to vaccinate is particularly high in pregnant or reproductive-aged women as multiple studies have shown. It would be important to comment on that given how serious COVID-19 infection is in pregnancy. 

Heightened vaccine hesitancy in women has been established in other studies, including the review article by Wang & Li that we now cite. Sex/gender was not our main focus here but we have updated the literature we cite in regards to this finding throughout.

Free text responses - the authors rightly acknowledge that they could not do proper qualitative analyses on these. I believe it is not right to impose a quantitative framework on a qualitative response. It does not matter what percentage of people mentioned lack of trust it is the depth, richness, and complexity are would be uncovered through qualitative responses. 

Thank you for this comment. We believe our analysis is appropriate to the type of data we have. In terms of presentation, we think there is value in providing the reader a sense between a much-mentioned and a once-mentioned sentiment. Nevertheless, we have greatly revised this section to emphasize the common themes that emerged and put the Figures in the Supplemental materials for those who want to know more about how we arrived at our conclusions. We have greatly revised out qualitative results section. See p. 24-26.

Discussion:

The limitations section is adequate. 

The authors say that higher vaccine hesitancy among Non-Hispanic Blacks is not entirely accounted for institutional mistrust compared to NH Whites Trump votes. However, it would be important to discuss from existing literature other possible reasons for this.

We have elaborated this point further in the Discussion on p. 19 as follows:

“Unlike vote choice, the significance of race and ethnicity persisted even after the inclusion of measures of mistrust, alternative media use, and individualism. It is possible that the particular measures of mistrust were insufficient to capture the nuanced mechanisms through which distrust in these institutions’ shapes COVID-19 vaccine attitudes. Prior studies have found that mistrust among Black populations in the U.S. may be further exacerbated by the general sociopolitical climate in the US [53], or potential confounding with socioeconomic status that is difficult to measure accurately and adjust for in observational studies with online samples (e.g., lack of financial resources or housing insecurity) [54].”

---

## [Decision Letter · Decision Letter 1]

21 Jun 2022

PONE-D-21-32376R1

Does Highlighting COVID-19 Disparities Reduce or Increase Vaccine Intentions? Evidence from a Survey Experiment in a Diverse Sample in New York State Prior to Vaccine Roll-out

PLOS ONE

Dear Dr. Fox,

Thank you for submitting your manuscript to PLOS ONE. After careful consideration, we have decided that your manuscript does not meet our criteria for publication and must therefore be rejected. Please see the reviewer's comments below, in particular Reviewer 1. 

I am very sorry that we cannot be more positive on this occasion, but hope that you appreciate the reasons for this decision.

Kind regards,

Andrew G Wu

Academic Editor

PLOS ONE

Reviewers' comments:

Reviewer's Responses to Questions

**Comments to the Author**

1. If the authors have adequately addressed your comments raised in a previous round of review and you feel that this manuscript is now acceptable for publication, you may indicate that here to bypass the “Comments to the Author” section, enter your conflict of interest statement in the “Confidential to Editor” section, and submit your "Accept" recommendation.

Reviewer #1: All comments have been addressed

Reviewer #2: All comments have been addressed

2. Is the manuscript technically sound, and do the data support the conclusions?

Reviewer #1: No

Reviewer #2: Yes

3. Has the statistical analysis been performed appropriately and rigorously? 

Reviewer #1: No

Reviewer #2: I Don't Know

4. Have the authors made all data underlying the findings in their manuscript fully available?

Reviewer #1: Yes

Reviewer #2: No

5. Is the manuscript presented in an intelligible fashion and written in standard English?

Reviewer #1: Yes

Reviewer #2: Yes

6. Review Comments to the Author

Reviewer #1: In the revised manuscript, the authors have reduced the scope of their claims, as requested. But I find the revised version of the manuscript more problematic than the original.

(1) The authors want to focus the new manuscript primarily on the experiment. The main research question stated is “does experimental exposure to a news report favoring minority prioritization of the vaccine due to the disproportionate burden of COVID-19 on these groups affect intent-to-vaccinate? If so, does the impact of reading this news report vary along race-ethnic and partisan lines?”

In this case we would need to scrutinize the experiment even more, and I’m not convinced that the experimental design can effectively isolate the effect of highlighting COVID-19 disparities. We must keep in mind that if the study is published, then readers (and the press) will get the impression that “highlighting disparities does not affect vaccine intentions” and I’m not sure this experiment convincingly demonstrates that.

(a) The messages in both treatment and control settings are focused on the prioritization of high-risk healthcare workers for vaccines. The treatment group focuses on “low income minority groups,” not racial minorities in general. Then the control group provides additional categories of health risks (the elderly, those with health conditions like obesity etc). There are too many things varying across conditions. The treatment neither adequately isolates the focus (i.e., just race-ethnicity) nor is identical to the control in other respects.

(b) The point I made last time that the images chosen (especially the white doctor and black patient) could inadvertently reinforce racist stereotypes and reduce trust may be a confounding factor at work.

(c) It seems that they should only be comparing cases which correctly comprehended the treatment and control messages – I’m not sure from reading the paper whether they analyzed the full sample or only those who correctly comprehended the message of each condition. If they’re analyzing the full sample then it seems to dilute the effect of the intervention.

For these reasons I’m not sufficiently convinced that the experiment is able to demonstrate the claim that highlighting disparities does not affect vaccine intentions. A differently designed experiment without some of these problems might produce different results.

(2) Even if someone were to find the experimental treatment convincing for isolating the effect of highlighting inequality, I don’t understand why the qualitative data were not analyzed by treatment vs experimental conditions. The choice to analyze by the 5Cs framework seems unrelated to the experiment – and there seems to be nothing in the qualitative data related to the new focus of the article (“highlighting COVID-19 disparities”)

(3) The third research question (association of race-ethnicity and party with intent-to-vaccinate) also seems to be unrelated to the new focus of the article and are not from a representative sample, so I don’t understand why this question is still in the paper.

Reviewer #2: Thank you for undertaking the revisions. I believe the points I raised have been addressed. I have no further comments.

7. PLOS authors have the option to publish the peer review history of their article (what does this mean?). If published, this will include your full peer review and any attached files.

Reviewer #1: No

Reviewer #2: No

- - - - -

---

## [Author Response · Author response to Decision Letter 1]

4 Aug 2022

Re: PLOS ONE Decision: PONE-D-21-32376R1 

July 5, 2022

Dear Editors,

We are writing to submit an appeal of the decision to reject our manuscript entitled “Does Highlighting COVID-19 Disparities Reduce or Increase Vaccine Intentions? Evidence from a Survey Experiment in a Diverse Sample in New York State Prior to Vaccine Roll-out.” Thank you for giving us the opportunity to appeal the decision to reject the manuscript.

We would first like to give an overview of why we are appealing the rejection, followed by a detailed response to Reviewer 1’s comment who found new concerns with the manuscript after we revised it.

There are two primary reasons that we are appealing the decision to reject the manuscript:

1. First, the reviewers were split with Reviewer 2 finding no further issues with the paper and Reviewer 1 raising some continued concerns. Thus, we feel the decision could have gone either way. We understand erring on the side of caution, but our second point outlines why we think the decision should go in the direction of accept.

2. Second, overall, we feel that the rejection is based on the fact that the experiment found null results and concern about how that might be interpreted by the media/academics, etc. We understand Reviewer 1’s concerns. However, we submitted to PLoS ONE precisely because our understanding is that journals like PLoS ONE are designed to address the so-called “file-drawer” problem whereby null results tend to be shoved in a file drawer. This can falsely skew science if only significant results in hypothesized directions are published. We do not dispute some of the limitations that Reviewer 1 raises about the experiment. However, we carefully outlined what these limitations are and provided sufficient information so that researchers looking to try similar experiments could improve on our design. 

We are also confident that our results, at least as the experiment was carried out, produced true null findings. Adding a message about racial equity to a message about prioritization for COVID-19 vaccines did not move the needle much on minority respondents’ intention to vaccinate and also did not move the needle on white respondents’ vaccine intentions. 

This article was initially well-received, having been presented at a number of prominent conferences. If the article is not published here, in all likelihood it will go on one or more of our websites and may never be officially published in a peer reviewed journal. However, we believe the academic community is better served by having the article published in a journal rather than not and that other researchers could learn from and improve upon approach. 

Below we describe in more detail our responses to Reviewer 1’s ongoing concerns and, where we were able to address their concerns, how we addressed them. Any changes made to the manuscript beyond the original revisions have been made in red in the clean manuscript.

Kind regards,

Ashley

-- 

Ashley M Fox, PhD, MA

Associate Professor

Department of Public Administration and Policy

Rockefeller College of Public Affairs and Policy

University at Albany, SUNY

Address: Milne 300C, 135 Western Avenue, Albany, NY, 12222

Tel: 518-442-5205

Website: https://www.albany.edu/rockefeller/faculty/ashley-m-fox

Response to Reviewers

Reviewer #1: In the revised manuscript, the authors have reduced the scope of their claims, as requested. But I find the revised version of the manuscript more problematic than the original.

(1) The authors want to focus the new manuscript primarily on the experiment. The main research question stated is “does experimental exposure to a news report favoring minority prioritization of the vaccine due to the disproportionate burden of COVID-19 on these groups affect intent-to-vaccinate? If so, does the impact of reading this news report vary along race-ethnic and partisan lines?”

In this case we would need to scrutinize the experiment even more, and I’m not convinced that the experimental design can effectively isolate the effect of highlighting COVID-19 disparities. We must keep in mind that if the study is published, then readers (and the press) will get the impression that “highlighting disparities does not affect vaccine intentions” and I’m not sure this experiment convincingly demonstrates that.

Response: See our overarching comment above about the “file drawer” problem. We understand the Reviewer’s concern. However, we are up front in the paper about the limitations of the study and we are also confident that the experiment, as performed, produced true null findings. This is not to say that improved messaging might not move the needle more, but we think that our approach was rigorous and others can improve upon it. We also think it is important to highlight that the disparities message did not reduce White respondents’ intention to vaccinate, which is a finding we also highlight.

(a) The messages in both treatment and control settings are focused on the prioritization of high-risk healthcare workers for vaccines. The treatment group focuses on “low income minority groups,” not racial minorities in general. Then the control group provides additional categories of health risks (the elderly, those with health conditions like obesity etc). There are too many things varying across conditions. The treatment neither adequately isolates the focus (i.e., just race-ethnicity) nor is identical to the control in other respects.

Response: We understand the Reviewer’s concern about the differences between the control and the experimental conditions. However, the overall message conveyed in both is quite consistent with the primary difference being that the experimental condition discusses prioritization of minorities and acknowledges racism is the root cause of health disparities. The title of the articles is identical and is focused on who should get priority for the vaccines when initial supply was limited. Both arms cite the recommendations of the U.S. Advisory Panel (https://www.nap.edu/catalog/25917/framework-for-equitable-allocation-of-covid-19-vaccine). The control arm goes on to elaborate about the recommendations and mentions nothing about minority prioritization. 

We also believe that the experimental arm does capture well the focus on race/ethnicity, particularly race. The experimental arm mentions a plea to vaccinate low-income minorities as well as a message acknowledging that racism is at the heart of COVID-19 related disparities in mortality. As we discuss in the paper, acknowledging that mistrust may be justified due to past and present injustices is a recommendation of heath communication scholars and what we were trying to capture in the experimental condition. 

Both articles were based off of actual news articles that we adapted for the experiment and therefore are realistic in terms of real world content the public might be exposed to. Both conditions are included at the end of this memo. We have highlighted in red font the parts of the conditions that differ. 

One reason for the null results might be that we had no “true” control condition that did not include any exposure to a news article about prioritization given our limited budget. We have now highlighted this in the paper as an additional potential reason for the null findings. This may allay some of the Reviewer’s concerns that this could be interpreted as finding that highlighting disparities does not allay vaccine hesitancy. 

Comment:

(b) The point I made last time that the images chosen (especially the white doctor and black patient) could inadvertently reinforce racist stereotypes and reduce trust may be a confounding factor at work.

Response: We did not dispute this point in the Revision nor do we dispute it now. We acknowledge that this is a limitation on p. 30-31.

However, we will also note that this recent paper by Gadarian et al (2022) shows that information from same-race/ethnicity experts online does not increase vaccine interest or intention to vaccinate, so it is also not clear that had the image been concordant that it would have changed the null findings. We have now added this paper to our discussion on p. 30. 

• Gadarian SK, Goodman SW, Michener J, Nyhan B, Pepinsky TB. Information From Same-Race/Ethnicity Experts Online Does Not Increase Vaccine Interest or Intention to Vaccinate. Milbank Q. 2022 Jun;100(2):492-503. doi: 10.1111/1468-0009.12561. Epub 2022 Mar 22. PMID: 35315950; PMCID: PMC9111148.

(c) It seems that they should only be comparing cases which correctly comprehended the treatment and control messages – I’m not sure from reading the paper whether they analyzed the full sample or only those who correctly comprehended the message of each condition. If they’re analyzing the full sample then it seems to dilute the effect of the intervention.

Response:

We understand the Reviewer’s point. To answer the Reviewer’s question, we are analyzing the full sample. However, to clarify, no one who is included in the full sample failed this attention check. We considered as acceptable to continue in the survey either those who answered that high-risk health care workers/first responders or low-income minorities should be prioritized for vaccines because the experimental prompt emphasized both. However, a total of 61 also included category #3 (Healthy Children and Young Adults) as being prioritized in addition to 1 and/or 2. We chose to retain these as they passed other checks (e.g., speeding, etc). We have rerun the analysis excluding those individuals and do not find any difference in the experimental outcome from excluding them. We have now added some more details to the paper about those who “failed” the test and the analyses taking out this group in the Supplemental Analyses. See Tables below.

S3 Table 1: Summary of Attention Check Responses to Experimental Condition

S3 Table 2: Experiment Interacted w/Race & Ethnicity, respondents failing attention check excluded

VARIABLES+A3:B45 odds ratio

Hispanic (ref) ref

NH White 0.99

 (0.684 - 1.432)

NH Black 1.55**

 (1.109 - 2.180)

Experimental Arm 1.27

 (0.898 - 1.809)

NH White#Experimental Arm 0.76

 (0.458 - 1.261)

NH Black#Experimental Arm 0.7

 (0.428 - 1.141)

Observations 1,290

NOTES: 

ciEform in parentheses 

*** p<0.01, ** p<0.05, * p<0.1 

 Gender, Income, Education and Age included but not shown

S3 Table 3: Experiment Interacted w/Vote Choice, respondents failing attention check excluded

VARIABLES odds ratio

Vote Choice Biden (2020) (ref) ref

Vote Choice Trump 2020 1.99***

 (1.299 - 3.048)

Vote Choice Other/No Vote 2020 1.65***

 (1.176 - 2.312)

Experimental Arm 1.2

 (0.920 - 1.552)

Vote Trump#Experiment 0.64

 (0.348 - 1.183)

Vote Other#Experiment 0.72

 (0.443 - 1.158)

Observations 1,290

Notes: Ordered Logistic Regression 

ci Eform in parentheses 

*** p<0.01, ** p<0.05, * p<0.1 

Gender, Income, Education and Age included but not shown

For these reasons I’m not sufficiently convinced that the experiment is able to demonstrate the claim that highlighting disparities does not affect vaccine intentions. A differently designed experiment without some of these problems might produce different results.

We do not dispute that a differently designed study might have produced different findings. Any study, experimental or otherwise, if designed differently, could produce different findings. We were careful to document what our hypotheses were, what our methods were, and what we found. Our design is not out of step with the methods of other experimental studies that we cite. We are honest about the limitations.

(2) Even if someone were to find the experimental treatment convincing for isolating the effect of highlighting inequality, I don’t understand why the qualitative data were not analyzed by treatment vs experimental conditions. The choice to analyze by the 5Cs framework seems unrelated to the experiment – and there seems to be nothing in the qualitative data related to the new focus of the article (“highlighting COVID-19 disparities”)

Response: First, let us clarify that the purpose of the qualitative results was not to examine whether there were differences in responses between treatment and control but rather to see whether we could derive qualitative insights into how different groups reason about their vaccine intentions. Thus, we did not try checking for differences in thematic expressions between treatment and controls. We suspect there would not be since we did find many references to racial disparities in the reasons given for people’s vaccination intentions. We do in fact mention this in the paper on p. 25 that we found a small (n=8) subset of non-Hispanic Black respondents explicitly stated not wanting to “be a guinea pig” or a “lab rat,” which we felt was indicative of concerns about medical experimentation on minorities. We used the 5 Cs because that is what emerged thematically from our coding. 

We found largely that the qualitative reasons offered for vaccine intentions did not differ much across racial, ethnic or ideological categorizations. This is a surprising finding, that we think is quite valuable for policymakers to know that the qualitative reasons given do not vary substantially even if perhaps the root causes of hesitancy do. 

(3) The third research question (association of race-ethnicity and party with intent-to-vaccinate) also seems to be unrelated to the new focus of the article and are not from a representative sample, so I don’t understand why this question is still in the paper.

Response: We believe these descriptive results from the sample are still valuable given the oversampling of racial and ethnic minorities, which allow for sub-analyses and the data are from a large sample of New Yorkers at a critical moment in the pandemic. We do not claim to be generalizing to the population, but the findings are in line with other studies reviewed in our literature review. Again, we believe in open-science and will make our dataset available to others upon publication if they want to try to apply sampling weights and make the data representative of New Yorkers.

Reviewer #2: Thank you for undertaking the revisions. I believe the points I raised have been addressed. I have no further comments.

---

## [Decision Letter · Decision Letter 2]

3 Oct 2022

PONE-D-21-32376R2

Does Highlighting COVID-19 Disparities Reduce or Increase Vaccine Intentions? Evidence from a Survey Experiment in a Diverse Sample in New York State Prior to Vaccine Roll-out

PLOS ONE

Dear Dr. Fox,

Thank you for submitting your manuscript to PLOS ONE. After careful consideration, we feel that it has merit but does not fully meet PLOS ONE’s publication criteria as it currently stands. Therefore, we invite you to submit a revised version of the manuscript that addresses the points raised during the review process.

Thank you for your appeal.  Both reviewers have responded to your appeal and your revised manuscript. 

The main concerns as stated by reviewer 1 pertain to the fact that this experiment did not have a true control, which therefore limits the interpretation of which can be taken by the findings.  As such you will need to ensure that reviewer 1s concerns are adequately addressed and acknowledged in the limitations section of the manuscript.  For example, in the revised submission I do not see in the limitations any consideration of the concerns raised i.e., that the treatment group is focused on ‘low-income minority groups’ not racial minorities in general.  The control also includes additional categories e.g., health risk (the elderly, those with health conditions like obesity etc) which again does not isolate the effect of race-ethnicity.  These are important factors which may have influenced the findings which should be acknowledged.  On that basis I invite you to resubmit a revised manuscript with a strengthened limitation which addresses the concerns of reviewer 1.

We look forward to receiving your revised manuscript.

Kind regards,

Erica Jane Cook, Ph.D

Academic Editor

PLOS ONE

Journal Requirements:

Additional Editor Comments (if provided):

Thank you for your appeal. Both reviewers have responded to your appeal and your revised manuscript.

The main concerns as stated by reviewer 1 (of which I agree) pertain to the fact that this experiment did not have a true control, which therefore limits the interpretation of which can be taken by the findings. As such you will need to ensure that reviewer 1s concerns are adequately addressed and acknowledged in the limitations section of the manuscript. For example, in the revised submission I do not see in the limitations any consideration of the concerns raised i.e., that the treatment group is focused on ‘low-income minority groups’ not racial minorities in general. The control also includes additional categories e.g., health risk (the elderly, those with health conditions like obesity etc) which again does not isolate the effect of race-ethnicity. These are important factors which may have influenced the findings which should be acknowledged.

On that basis I invite you to resubmit a revised manuscript with a strengthened limitation which addresses the concerns of reviewer 1.

Reviewers' comments:

Reviewer's Responses to Questions

**Comments to the Author**

1. If the authors have adequately addressed your comments raised in a previous round of review and you feel that this manuscript is now acceptable for publication, you may indicate that here to bypass the “Comments to the Author” section, enter your conflict of interest statement in the “Confidential to Editor” section, and submit your "Accept" recommendation.

Reviewer #1: All comments have been addressed

Reviewer #2: (No Response)

2. Is the manuscript technically sound, and do the data support the conclusions?

Reviewer #1: No

Reviewer #2: Partly

3. Has the statistical analysis been performed appropriately and rigorously? 

Reviewer #1: Yes

Reviewer #2: I Don't Know

4. Have the authors made all data underlying the findings in their manuscript fully available?

Reviewer #1: Yes

Reviewer #2: Yes

5. Is the manuscript presented in an intelligible fashion and written in standard English?

Reviewer #1: Yes

Reviewer #2: Yes

6. Review Comments to the Author

Reviewer #1: Having read the authors' appeal, I am still not convinced by their arguments:

(1) The "file drawer" problem is not relevant here. The reason I did not recommend publication was not because of the null finding; my concern is about what I find to be a problematic experiment. I am not convinced that the experiment has sufficient internal validity, and am concerned that the tendency of media/readers (whom the journal cannot control) to draw broader conclusions from this study (because of its publication in a highly prestigious journal) could be damaging.

(2) Because of the number of possible confounds in the treatment vs. control settings and lack of a true control, I can't trust the finding as a true null. Let me restate my concerns, which I don't find sufficiently addressed: "The treatment group focuses on “low income minority groups,” not racial minorities in general. Then the control group provides additional categories of health risks (the elderly, those with health conditions like obesity etc). There are too many things varying across conditions. The treatment neither adequately isolates the focus (i.e., just race-ethnicity) nor is identical to the control in other respects." I just don't think we know how respondents are interpreting the messages. The treatment setting is mixing race and class, and to conclude from this that the experiment isolates the effect of race is problematic. (E.g., if I were an African American individual reading the treatment messaging, I might read it as saying that the vaccine is meant primarily for lower-income individuals). Without cognitive interviews, who knows how people are reading these messages?

The fact that other scholars in conferences where the authors have presented this study do not see any problems with the experiment is baffling to me. The editor is welcome to recruit a third reviewer.

Reviewer #2: Thank you for the opportunity to read this manuscript again. I have read it again and note that the limitations section has been revised by the authors.

7. PLOS authors have the option to publish the peer review history of their article (what does this mean?). If published, this will include your full peer review and any attached files.

Reviewer #1: No

Reviewer #2: No

---

## [Author Response · Author response to Decision Letter 2]

9 Oct 2022

Re: PLOS ONE Decision: PONE-D-21-32376R1 

October 9, 2022

Dear Editors,

Thank you for considering our appeal of the decision to reject our manuscript entitled “Does Highlighting COVID-19 Disparities Reduce or Increase Vaccine Intentions? Evidence from a Survey Experiment in a Diverse Sample in New York State Prior to Vaccine Roll-out.” We appreciate the Editor giving us the opportunity to further revise our paper. 

In response to the remaining concern of the Reviewer and Editor that the experiment did not have a true control, and that this was not adequately discussed in the Limitations section, we have added the following paragraph to the Limitations on p. 32: 

“A final limitation is that we did not include a “true” control condition that excludes any exposure to a news article about prioritization given our limited budget. We therefore cannot say for certain whether respondents’ intent to vaccinate would have looked different if they had not seen any article at the outset of the survey. For the purpose of this paper, we have treated the condition about prioritization of health workers and other priority groups (but without any reference to race or ethnicity) as the control. While the overall message conveyed in both is quite similar as both arms cite the recommendations of the U.S. Vaccine Advisory Panel, there are several differences in the wording and listing of groups being prioritized between the two conditions. To view the full set of ways that the treatment and control arms differ in wording, the full treatment and control arms are included in the Supplementary Materials. However, the title of the articles is identical and is focused on who should get priority for the vaccines when initial supply was limited with the primary difference being that the experimental condition discusses prioritization of low-income minorities and acknowledges racism is the root cause of health disparities. Both articles are adapted from actual news stories and therefore are realistic exposures that the public might consume.”

We hope this is sufficient to allay the continued concerns about the paper. 

To the point about the difficulties disentangling race and class due to the reference to “low-income” minorities, first, we will note that this was the wording used in the actual news article we selected and not our own wording. It is true that we could have taken out the reference to low-income. However, in the U.S. context we believe that the two are so entangled that attempts to disentangle them are fraught. Even if we had not included the phrase ‘low-income,’ we might hypothesize that this would be how the term “minorities” would be interpreted by respondents.

Please feel free to reach out with any continued concerns.

Kind regards,

Ashley

-- 

Ashley M Fox, PhD, MA

Associate Professor

Department of Public Administration and Policy

Rockefeller College of Public Affairs and Policy

University at Albany, SUNY

Address: Milne 300C, 135 Western Avenue, Albany, NY, 12222

Tel: 518-442-5205

Website: https://www.albany.edu/rockefeller/faculty/ashley-m-fox

---

## [Editor Report · Decision Letter 3]

19 Oct 2022

Does Highlighting COVID-19 Disparities Reduce or Increase Vaccine Intentions? Evidence from a Survey Experiment in a Diverse Sample in New York State Prior to Vaccine Roll-out

PONE-D-21-32376R3

Dear Dr. Fox,

We’re pleased to inform you that your manuscript has been judged scientifically suitable for publication and will be formally accepted for publication once it meets all outstanding technical requirements.

Kind regards,

Erica Jane Cook, Ph.D

Academic Editor

PLOS ONE
---

## [Editor Report · Acceptance letter]

21 Nov 2022

PONE-D-21-32376R3 

Does Highlighting COVID-19 Disparities Reduce or Increase Vaccine Intentions? Evidence from a Survey Experiment in a Diverse Sample in New York State Prior to Vaccine Roll-out 

Dear Dr. Fox:

I'm pleased to inform you that your manuscript has been deemed suitable for publication in PLOS ONE. Congratulations! Your manuscript is now with our production department. 

Kind regards, 

on behalf of

Dr. Erica Jane Cook 

Academic Editor

PLOS ONE